# Biomimetic Material for Quantification of Methotrexate Using Sensor Based on Molecularly Imprinted Polypyrrole Film and MWCNT/GCE

**DOI:** 10.3390/biomimetics8010077

**Published:** 2023-02-12

**Authors:** Eduardo Jara-Cornejo, Sabir Khan, Jaime Vega-Chacón, Ademar Wong, Lariel Chagas da Silva Neres, Gino Picasso, Maria D. P. T. Sotomayor

**Affiliations:** 1Laboratory of Physical Chemistry Research, Faculty of Sciences, National University of Engineering, Av. Tupac Amaru 210, Lima 15333, Peru; 2Institute of Chemistry, São Paulo State University (UNESP), Araraquara 14801-970, Brazil; 3Department of Natural Sciences, Mathematics, and Statistics, Federal Rural University of the Semi−Arid, Mossoró 59625-900, Brazil; 4National Institute of Alternative Technologies for Detection, Toxicological Evaluation and Removal of Micropollutants and Radioactive Agents (INCT−DATREM), Araraquara 14801-970, Brazil

**Keywords:** molecularly imprinted polymer, polypyrrole, methotrexate, biomimetic sensors, sensor film, electropolymerization, MWCNT

## Abstract

This study investigates biomimetic sensors for the detection of methotrexate contaminants in environmental samples. Sensors inspired by biological systems are the focus of this biomimetic strategy. Methotrexate is an antimetabolite that is widely used for the treatment of cancer and autoimmune diseases. Due to the widespread use of methotrexate and its rampant disposal into the environment, the residues of this drug are regarded as an emerging contaminant of huge concern, considering that exposure to the contaminant has been found to lead to the inhibition of some essential metabolic processes, posing serious risks to humans and other living beings. In this context, this work aims to quantify methotrexate through the application of a highly efficient biomimetic electrochemical sensor constructed using polypyrrole−based molecularly imprinted polymer (MIP) electrodeposited by cyclic voltammetry on a glassy carbon electrode (GCE) modified with multi−walled carbon nanotubes (MWCNT). The electrodeposited polymeric films were characterized by infrared spectrometry (FTIR), scanning electron microscopy (SEM), and cyclic voltammetry (CV). The analyses conducted using differential pulse voltammetry (DPV) yielded a detection limit of 2.7 × 10^−9^ mol L^−1^ for methotrexate, a linear range of 0.01–125 μmol L^−1^, and a sensitivity of 0.152 μA L mol^−1^. The results obtained from the analysis of the selectivity of the proposed sensor through the incorporation of interferents in the standard solution pointed to an electrochemical signal decay of only 15.4%. The findings of this study show that the proposed sensor is highly promising and suitable for use in the quantification of methotrexate in environmental samples.

## 1. Introduction

Methotrexate (MTX) is an antimetabolite of the folate group that is widely used for cancer treatments in oncology hospital units worldwide; due to its widespread use, the residues of the drug are usually disposed of in hospital effluents [1]. MTX is given in high doses for the treatment of various types of cancer, including leukemia [2], prostate, and bladder cancer [3], and in low doses for the treatment of psoriasis, idiopathic arthritis, and juvenile dermatomyositis [4], and for photodynamic therapy [5], among other treatment procedures. However, as the difference between the minimum effective concentration and the minimum toxic concentration of methotrexate is small, the drug application needs to be strictly controlled in order to reduce latent toxicity [4].

Owing to its antimetabolite properties, exposure to MTX has been found to inhibit metabolic processes in the organism, and this poses serious risks to the health of humans and other living beings; in addition, due to the widespread use of methotrexate and its rampant disposal into the environment, the residues of this drug are regarded as an emerging contaminant of huge concern worldwide [6,7]. Bearing this in mind, over the last decade, a considerable amount of research has been focused on quantifying methotrexate in wastewater from oncology hospital units and in effluents from areas close to pharmaceutical industries and drugstores [8,9]. When it comes to dealing with contaminants of emerging concern, rigorous detection techniques and quantification control measures are required in order to avoid or mitigate the deleterious contamination effects of these compounds. The field of biomimetic materials is a creative branch of science that takes cues from the natural world to create beneficial products for humans [10]. The term “proof of concept” is often used to describe biomimicry, which is the process of adopting and adapting solutions found in nature [11]. Protecting human health, the environment, and national security all depend on the ability to detect specific target compounds with high sensitivity and selectivity. The development of highly precise and targeted sensors has been greatly influenced by observations made in nature. Sensors inspired by biological systems are the focus of this biomimetic strategy. There are many different ways in which biomimetic sensors can be fashioned after biological systems, including their functionality or principle, shape, strategy or behavior, or manufacturing [12].

A wide range of detection and quantification techniques have been developed targeted at determining the presence of contaminants of emerging concern in different matrices; among these techniques include high−performance liquid chromatography equipped with UV−visible detection [13], liquid chromatography−mass spectroscopy [9,14], UV−visible spectrophotometry [15], and electrochemical methods [16,17]. As widely reported in the literature, although most of the analytical methods are found to be relatively efficient for the treatment of biological matrices, such as blood [18,19] and urine [16,20], the efficiency of these methods is found to be compromised when they are applied for the treatment of contaminants in aquatic matrices [18,19] because of the lower concentration of the compounds in water.

To develop an effective biomimetic sensor coupled with smart material, it is essential to choose the specific junction between the molecular recognition cavity and complementary chemical groups within three−dimensional structures [21]. For effective sensing responses, recognition elements such as DNA/RNA aptamers and oligopeptides must be immobilized or coated on a transducing substrate before their use. Some methods for the detection of methotrexate, such as potentiometric, voltammetric, and impedance, have been used with selected biomimetic materials [22]. For example, artificial lipid polymer membranes and chalcogenide glass with PVC (Polyvinyl chloride) film are two examples of biomimetic materials utilized in sensors [23]. There can be more variation in analyte quantification and signal transduction with the use of bioengineered cells. The biomimetic sensors can be used to detect contaminants in food, drugs, and environmental samples [24,25,26].

Considering that the quantification of methotrexate concentration in aquatic matrices has generally been performed using highly expensive analytical methods, the present study aims to develop a highly efficient molecularly imprinted polymer (MIP)−based technique for methotrexate quantification, which is of low cost and involves simple preparation, in addition to exhibiting excellent stability and analytical specificity [27]. Molecularly imprinted polymers are derived from the polymerization of a functional monomer in the presence of a template molecule (generally the analyte); in order to obtain a highly efficient MIP, there needs to be a suitable affinity between the functional monomer and the analyte [28], since polymerization is expected to occur around the analyte. In this way, once the analyte is removed, free cavities are generated within the polymer, and these cavities are rendered fundamentally selective for the analyte [27,28,29]. This material obtained can be considered a biomimetic material that performs the same function as a biological system (enzyme and antibody) [30].

Several studies reported in the literature have successfully demonstrated the use of MIPs for the quantification of large molecules such as proteins and drugs [31,32,33] since these analytes could enter into the cavities of the imprinted polymer. However, it is worth noting that sensors based on only MIP generally do not have as suitable conductivity [34]. In this sense, the modification of the electrode surface with some conductive material can improve the electrical properties allowing for analytes detection at low limits of detection. One of these conductive substrates commonly used in sensors is the multi−walled carbon nanotube (MWCNT); the use of this material contributes effectively to accelerating the transfer of electrons and improving the sensitivity of sensors based on MIP [35]. Polypyrrole (PPy) is a highly conductive polymer that has been successfully applied in the construction of MIP−based sensors [36,37,38,39]. The combined application of PPy and MWCNT produces a synergistic effect in electrochemical detection analyses, favoring the effective detection of analytes [40].

The use of the electrochemical technique for the synthesis of polypyrrole has gained enormous traction among researchers because electropolymerization is a simple and straightforward process that allows the quantification of molecules of small and large molecular mass of the analyte even in minute concentration [36,37,38,39].

A wide range of electrochemical methods, including cyclic voltammetry [41], differential pulse [38], chronoamperometry [42], and electrochemical impedance [43], have been successfully employed in combination with molecularly imprinted polypyrrole−modified electrodes for the quantification of drugs in different matrices.

Taking the aforementioned considerations into account, this work reports the development and application of an electrochemical sensor based on molecularly imprinted polypyrrole electrodeposited on GCE modified with MWCNT for the quantification of methotrexate in real matrices (Figure 1); the proposed sensor exhibits excellent electrical conductivity and selectivity, which are mainly derived from the conductive properties of MWCNT and polypyrrole imprinted cavities incorporated into the sensing device. The applicability of the sensor was tested in pharmaceutical formulations and river water samples, and the proposed method was compared with the HPLC−UV method.

## 2. Materials and Methods

### 2.1. Instruments and Reagents

The morphological analysis of the molecularly imprinted polymers was conducted using scanning electron microscope model JSM 7500F. The functional groups of the molecularly imprinted polymer were evaluated using Bruker brand FTIR equipment. All the electrochemical measurements were performed using the conventional three−electrode system. The three−electrode system consisted of a glassy carbon electrode (d = 3 mm) as the working electrode, Ag/AgCl reference electrode (KCl 3 mol L^−1^), and an auxiliary platinum wire electrode. All the electrochemical measurements were performed at room temperature. The electrochemical measurements were conducted using AUTO LAB PGSTAT 100 potentiostat–galvanostat controlled by NOVA 2.1.1 software. All the reagents used for the experiments were of analytical grade, and all solutions were prepared in deionized water obtained from Milli−Q Direct−0.3 (Millipore).

Methotrexate (98%), pyrrole (99%), lithium perchlorate (98%), MWCNT (<5% modified with COOH), potassium hexacyanoferrate (II) trihydrate (98%), and potassium ferricyanide (98%) were purchased from Sigma−Aldrich. Sodium phosphate monobasic (98%), sodium phosphate dibasic (98%), and potassium chloride (98%) were acquired from Merck.

### 2.2. Modificacion of GCE with MWCNT

To improve the analytical signal of the sensor, a volume of 1 mg/mL of MWCNT was dispersed in N,N′-dimethylformamide; this dispersion was first sonicated in an ultrasound bath for 20 min and was subsequently deposited 7 µL of the mixture on the electrode surface. IR lamp was used for the complete evaporation of the solvent until drying.

### 2.3. Preparation of Molecularly Imprinted Polymer (MIP) and Molecularly Non−Imprinted Polymer (NIP) on MWCNT/GCE

The MIP/MWCNT/GCE was obtained by immersing the MWCNT−modified electrode in a solution containing 20 mmol L^−1^ pyrrole, 0.1 mol L^−1^ lithium perchlorate, and 100 µmol L^−1^ methotrexate; the NIP/MWCNT/GCE was prepared using the aforementioned solution but in the absence of methotrexate. The electropolymerization parameters employed were as follows: conditioning potential of 0.4 V for 30 s, potential range of −0.4 to 1 V, scan rate of 50 mV s^−1,^ and 25 electropolymerization cycles. After obtaining the electrodeposited polymer, we then proceeded with the analyte removal. For the analyte removal, the electrode was immersed in a sodium phosphate buffer solution of pH 10, and cyclic voltammetry analysis (8 cycles in the potential range of 0.3–1 V) was conducted to remove methotrexate from the imprinted polymer cavities. It was evidenced by the decrease in the oxidation peak of methotrexate.

### 2.4. Using the Electropolymerized Sensor for the Quantification of Methotrexate

The quantification of methotrexate was carried out using differential pulse voltammetry (DPV) with a pulse potential amplitude of 50 mV, pulse duration time of 50 ms, and a scan rate of 20 mV s^−1^. The electrode modified with polypyrrole was immersed in a standard solution of methotrexate dissolved in 0.1 mol L^−1^ sodium phosphate buffer solution and stirred for 2 min; the preconcentration of the analyte was conducted using an accumulation potential of −0.3 V for 30 s. After the preconcentration of methotrexate on the electrode surface, a potential of −0.1 to −0.9 V was applied in order to quantify the concentration of methotrexate using the DPV cathodic peak current.

## 3. Results and Discussions

### 3.1. Preparation of the GCE Modified with MIP/MWCNT and NIP/MWCNT

To obtain optimal conductive polypyrrole−based polymeric film, one needs to ensure suitable adhesion of the monomer to the substrate. MWCTN was employed as a substrate for the electropolymerization process; this material was chosen because it has been shown to improve the electrical properties of the sensor, in addition to providing strong adsorption and enhancing surface reaction activity [44]. An analysis was conducted in order to evaluate the optimal amount of MWCNT (1 mg/mL) to be added to the electrode surface (Appendix A); the result obtained from this analysis pointed to an optimal MWCNT concentration of 7 µL. Appendix A show that MWCNT/GCE recorded an anode current intensity 1.75 times greater than GCE due to the enhanced electrical properties derived from MWCNT.

Appendix A shows the process involving the electropolymerization of MIP/MWCNT/GCE and NIP/MWCNT/GCE; here, one can clearly observe the formation of polypyrrole film through the oxidation peak at +0.8 V during the electropolymerization of NIP/MWCNT/GCE [40]. In the case of the MIP sensor, methotrexate was used because it is the template molecule. In the case of the NIP, methotrexate is not used because it is desired to obtain a non−printed polymer in order to demonstrate the contribution of the printed cavities in the biomimetic sensors. In the first cycle of MIP/MWCNT/GCE electropolymerization, an increase in current was observed in the oxidation peak since both methotrexate and pyrrole were simultaneously oxidized at the same potential (of +0.8 V) [40,45]; this can be attributed to the presence of methotrexate in the printed cavities of the polymer, in the first instance oxidation occurs at 0.8 V but then during electropolymerization at a potential of −0.3 V it occurs that methotrexate is reduced and this cycle is repeated again at as the electropolymerization synthesis proceeds. Theoretically, this is possible due to the hydrogen bonding interactions that exist between the hydrogen of the N−H group of pyrroles and the OH and N−H groups of the methotrexate template molecule, which causes the molecule to not detach from the printed cavity. However, when the electrode is immersed at pH 10, it causes the methotrexate to obtain negative charges and through a cycling potential of 0.3 to 1 V that partially overoxidizes the polypyrrole, resulting in the sensor phase getting partially negative charges, therefore, through the repulsion of charges is achieved by the removal of methotrexate in the printed cavity.

Previous studies reported in the literature have shown that the electrical properties of the conductive polymer depend on the nature of the doping; the application of a suitable doping agent improves the electrical properties of the material, and this leads to a significant increase in the growth of the polymer during the electrodeposition synthesis. On the other hand, a doping agent that provides low conductivity to the polymer impeded the growth of the polymer [46]. In our present study, we employed perchlorate as the doping agent of polypyrrole; as expected, this agent provided suitable conductivity to the material. Another factor that favors the electrical properties of the sensor is the potential for prepolymerization. The next step is to optimize the electropolymerization potential range, Epoly, of the NIP/MWCNT/GCE polypyrrole film synthesized at pH 3. In Figure 2, the prepolymerization potential and the range of the electropolymerization potential are evaluated. The modified electrode was immersed in a solution containing 5 mmol L^−1^ [Fe(CN)_6_]^−3/−4^ and 0.1 mol L^−1^ KCl and it was analyzed by cyclic voltammetry at a scan rate of 50 mVs^−1^ to evaluate the electrical properties of the electrodeposited film in different conditions under the polymerization potential ranges; A ferricyanide solution is used, since this complex molecule has a negative charge and the electrodeposited polymer has positive charges, in this way it is very useful for us to evaluate these electrostatic interactions, the results obtained from this analysis showed that the current intensity of the oxidation of Fe^2+/3+^ is greater for the conditions potential range of −0.4 to 0.9 V with prepolymerization potential of 0.4 V yielded a better electrochemical response from the redox probe (Figure 2), this result shows a greater conductive property of the sensor; therefore, these synthesis conditions would favor the increase in the sensitivity of the sensor to detect methotrexate. The range from −0.4 to 0.9 V has a greater potential range during electropolymerization because a larger amount of polypyrrole is achieved on the surface of the modified electrode compared to the range 0 to 1 V. This greater amount of polypyrrole facilitated the electrical conductivity showing a higher intensity of ferricyanide current. Appendix A provides more details about this study.

### 3.2. Electrochemical Behavior of MTX on the Modified Electrode

The analysis involving the quantification of methotrexate was performed using conditioning time and conditioning potential; this essentially enabled us to obtain a stable current. Appendix A shows the cathodic current of methotrexate relative to different conditioning times and conditioning potentials. Looking at Appendix A, one will observe an increase in current up to 30 s, after which the current time remained constant because the electrode surface was saturated with methotrexate molecules. On the other hand, Appendix A shows that, at negative potentials, the electrode surface attracted a relatively higher amount of methotrexate molecules due to the electrostatic interaction between the negatively charged surface and the cationic amine group of methotrexates. The optimum conditioning potential was −0.3 V.

As shown in Figure 3, the results obtained from the DPV analyses show that MIP/MWCNT/GCE exhibits greater sensitivity compared to NIP/MWCNT/GCE; this is attributed to the imprinted cavities that were generated during the synthesis procedure. After that, the voltammetry profile of the MIP/MWCNT/GCE was evaluated; the results obtained from this analysis pointed to a potential shift of approximately 50 mV in methotrexate reduction on the surface of the modified electrode. This cathodic displacement can be attributed to the electrocatalytic effect of the electrodeposited polymer and the imprinted cavities on the modified electrode. In addition, we also observed a significant increase in the peak current when polypyrrole was used to modify the electrode; this can be attributed to the increase in the electroactive area caused by the application of this polymer. The values of the electroactive area were calculated using the Randles−Sevcik equation, i_p_ = 2.69 × 10^5^ n^3/2^AD^1/2^cv^1/2^, where n is the electrons involved in the reduction, A electroactive area, D diffusion constant of the electrolyte, C concentration of the electrolyte and v scan rate (Appendix A); the results obtained showed that GCE, MWCNT/GCE, NIP/MWCNT, and MIP/MWCNT/GCE exhibited electroactive area (in cm^2^) of 0.104, 0.157, 0.166 and 0.184, respectively. Polypyrrole is known to be a conducting polymer due to the presence of the delocalized π−electron system and the doping that occurs during synthesis. However, it is observed that the change in the morphology of the polymer produced a slight increase in its electroactive area. Figure 4 presents the possible mechanism involving the MTX reduction reaction [45].

### 3.3. Optimization of Experimental Parameters Related to the Electropolymerization and Characterization of the Proposed Polymeric Film

The number of electropolymerization cycles and concentration of pyrrole and methotrexate are relevant factors that need to be considered during the electropolymerization process since they are directly related to the thickness of the film, the physical stability of the polymer, and the formation of the cavities of the polymer [38]. For the sensor optimization analysis, the cathodic signal of methotrexate was quantified by immersing the modified electrode in a solution containing 0.1 mol L^−1^ phosphate buffer and 0.1 mol L^−1^ KCl at pH 3. Figure 5 shows that the optimal conditions of the experiment involved the following: electropolymerization in a potential range of −0.4 to 0.9 V for 25 cycles in 0.1 mol L^−1^ LiClO_4_ solution containing 20 × 10^−3^ mol L^−1^ pyrrole and 100 × 10^−6^ mol L^−1^ methotrexate.

The FTIR technique was used to characterize the functional groups of the imprinted polymer. In Figure 6b, the characteristic bands at 1250, 1450, and 1360 cm^−1^ in Figure 6 confirm the presence of PPy [47]. The frequencies at 1606 and 1452 cm reflect the C−N stretching vibration in the ring plane and CN strain modes, respectively. The presence of skeletal vibrations at 1606 cm^−1^ and strong bands at 732 cm^−1^ typically reflect a five−membered aromatic ring [47]. For OPPy/MWCNT, the N−C=O functional group was evidenced at the bands at 1700 cm^−1^. It is related to the methotrexate removal process since it is expected that the polymer partially overoxidizes; that is, it obtains some carbonyl groups in the scent ring.

The morphology of the (synthesized) polypyrrole for MIP/GCE (a), MWCNT/GCE (b), and MIP/MWCNT/GCE can be found in Figure 7. Due to the potential applied before electropolymerization, the pretreatment procedure stimulates the vertical growth of the electrodeposited polymer, where an average width of 77.1 ± 20.5 nm and a polydispersity (polyd. = 100% σ/$\overset{-}{x}$, where σ is standard deviation and $\overset{-}{x}$ is mean) value of 26.6 are obtained for MIP/GCE samples. These results help to confirm that polypyrrole nano tubules were obtained during the synthesis of MIP/GCE [48]. The vertical growth and the nanometric size of the polymer are seen to indicate an increase in electroactive sites. Based on the analysis of the voltammogram profiles of [Fe(CN)_6_]−3/−4 shown in Figure 2, we noted that the electrodeposited films under this prepolymerization potential exhibited an increase in electrochemical signal compared to that of the electrodeposited films that were not subjected to pretreatment. Thus, one can say that the application of oriented electrodeposition resulted in an increase in the electroactive sites of the sensor. In this sense, when it comes to the direct quantification of methotrexate, one expects that this electropolymerization pretreatment favors the number of bridging interactions—hydrogen between the polymer and analyte. Figure 7 shows the growth of polypyrrole around the substrate—MWCNT.

### 3.4. Analytical Performance

The study of the pH level of methotrexate reduction was conducted by DPV using a supporting electrolyte composed of 0.1 mol L^−1^ PBS and 0.1 mol L^−1^ KCl in the presence of 100 µmol L^−1^ methotrexate at pH 3, 5, 7, and 9, with the applied potential ranging from −0.3 to −0.9 V (Figure 8). The results obtained from this analysis showed that the electrochemical reduction in methotrexate is largely dependent on the pH value of the solution [45]. The electrochemical signal of the MIP/MWCNT/GCE sensor was found to be pH dependent with linearity of Epc = −0.045 pH—0.31, based on the application of PBS as a supporting electrolyte. This outcome points to the decrease in MTX adsorption at basic pH, which could be attributed to the first (pKa1 = 3.5) and second (pKa2 = 8.8) deprotonation of the carboxylic acid groups of MTX. Since an accumulation potential of −0.3 V was applied during the pretreatment, this caused the sensor surface to be negatively charged, and through the repulsion of charges, it repelled the deprotonated methotrexate molecules. Thus, the result showed that the maximum adsorption capacity of the electrodeposited polymer for methotrexate quantification was obtained at pH 3. In order to justify this pH choice, Appendix A shows the influence of the pH in the structure of the MTX molecule.

With regard to the MIP/MWCNT/GCE film, a calibration curve was obtained based on the application of the following linear regressions: ΔI_p1_ (μA) = 0.226 [MTX] + 0.35 with a correlation coefficient of 0.98 for a concentration range of 0.01–25 μmol L^−1^, and ΔI_p2_ (μA) = 0.113 [MTX] + 2.62 with a correlation coefficient of 0.995 for a concentration range of 25–125 μmol L^−1^. As can be observed in Figure 9, the presence of these two concentration ranges can be attributed to the affinity of methotrexate with the imprinted cavities of the polymer since, at low analyte concentrations, the molecules tend to be adsorbed by the cavities with high electrochemical affinity located on the external area of the polymer. At higher analyte concentrations, one observes a saturation of these high−affinity cavities due to the increase in the concentration of methotrexate; thus, the lower affinity cavities that are located deep in the polymer are responsible for adsorbing the analyte, causing a decrease in sensitivity [38].

To calculate the limit of detection (LOD), we employed the following equation: 3S_B_/m, where S_B_ is the standard deviation of 10 blank solution measurements and m corresponds to its slope [49]; the limit of quantification was calculated using the following equation: LQ = 10 S_B_/m. The values obtained for LOD and LOQ were 2.7 × 10^−9^ mol L^−1^ and 9 × 10^−9^ mol L^−1^, respectively.

### 3.5. Selectivity Study

A highly efficient MIP needs to be endowed with a high selectivity for the template molecule. Specific recognition is based on the interaction between the functional groups of the molecule of interest with the functional groups of the imprinted sites. Two crucial factors are known to be key when it comes to selectivity: the functional groups of the molecules and their size; thus, molecules that possess a similar structure tend to exhibit a greater interference effect. The selectivity of the sensor was evaluated in a binary solution (analyte and interferent) through a comparative analysis of the DPV response. To perform this analysis, the sensor was immersed in a solution containing methotrexate and the interferent at a concentration of 100 μmol L^−1^; the objective here was to quantify MTX in real samples containing molecules such as chloride, nitrates, sulfates, L−aspartic acid (L−AA), amoxicillin (AMX), tetracycline (TTC), and ivermectin (ITV).

As can be observed in Figure 10, AMX and TTC molecules exhibited an interference effect of 14.65% and 15.42%, respectively. This can be attributed to the size of AMX and TTC molecules, which is similar to that of the methotrexate molecule; also, AMX and TTC possess amine and OH groups, which are capable of binding through a hydrogen bond, and thus allow higher accessibility to the cavities. A larger molecule with a different structural shape, such as ivermectin, does not interfere with the analytical signal of MTX due to stereochemical obstacles; as a result, IVT exhibited a low interference effect of 3.6%. A numerical representation is used to quantify the contribution of the factor of molecularly imprinted cavities (α) and selectivity (β). This common strategy widely employed for the analysis of sensor selectivity allows us to confirm what was stated above relative to IVT; looking at Table 1, one will see that IVT exhibited the highest alpha and beta values, and this essentially shows that the imprinted cavities had a higher affinity with methotrexate. The analytical results obtained for all the molecules investigated are shown in Table 1.

### 3.6. Quantification of Methotrexate in Pharmaceutical and River Water Samples

A thorough analysis was conducted in order to test the applicability of the proposed sensor in real samples; this analysis was conducted using pharmaceutical formulations and river water samples. For the analysis of the applicability of the sensor in terms of MTX quantification in pharmaceutical samples, tablets of 2.5 mg MTX were used. The response of the sensor was also tested in river water samples and was used directly without pretreatment, with the addition of standard MTX of 2.5 and 25 μmol L^−1^. The results obtained showed that the proposed sensor exhibited excellent recovery values—see Table 2 and Table 3.

## 4. Conclusions

The present work reported the successful development and application of the MIP/MWCNT/GCE sensor for the quantification of methotrexate in environmental and pharmaceutical samples and, when compared with other sensors for this analyte, presented very suitable characteristics (Appendix A). The sensor coupled with smart biomimetic material of MIPs to obtain the selective recognition cavities for methotrexate through the electropolymerization of pyrrole in GCE modified with MWCNT. The MIP/MWCNT/GCE sensor having selective cavities showed suitable repeatability in the analysis of 5 successive measurements of methotrexate with a relative standard deviation (RSD) of 2.5% and a suitable reproducibility in five independent electrochemical measurements with RSD of 1.72%.

## Figures and Tables

**Figure 1 biomimetics-08-00077-f001:**
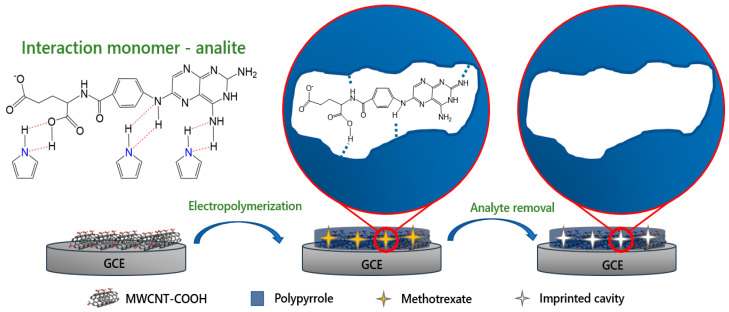
Schematic illustration of the preparation of electropolymerized MIP film and its application for the quantification of methotrexate.

**Figure 2 biomimetics-08-00077-f002:**
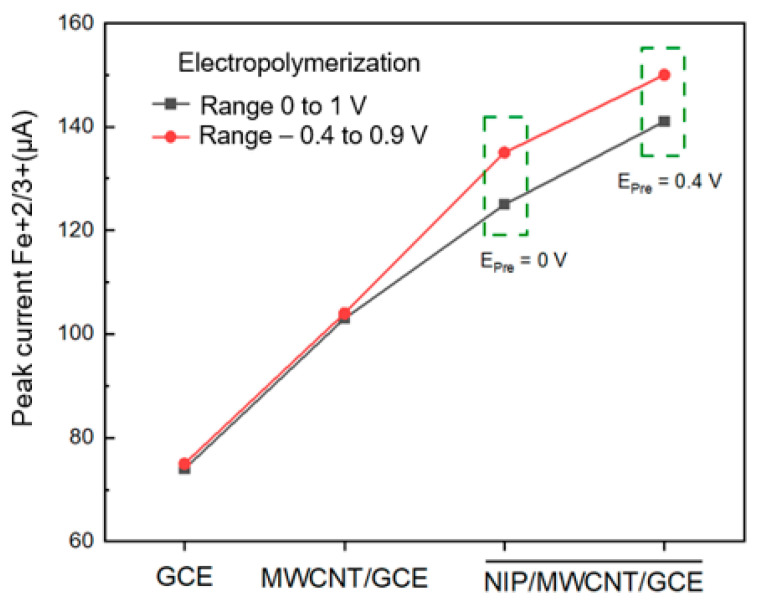
Comparison of current peaks Fe+2/+3 using GCE, MWCNT/GCE, and NIP/MWCNT/GCE sensor employed in the electropolymerization potential range of 0 to 1 V (black line) and −0.4 to 0.9 V (red line).

**Figure 3 biomimetics-08-00077-f003:**
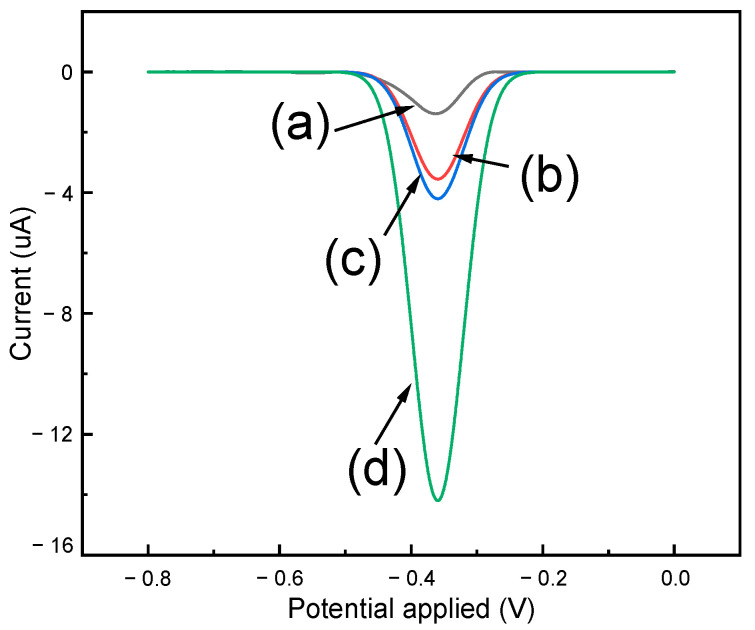
Differential pulse voltammograms obtained for (**a**) GCE, (**b**) MWCNT/GCE, (**c**) NIP/MWCNT/GCE, and (**d**) MIP/MWCNT/GCE applied in PBS (at pH 3) in the presence of 100 μmol L^−1^ methotrexate.

**Figure 4 biomimetics-08-00077-f004:**
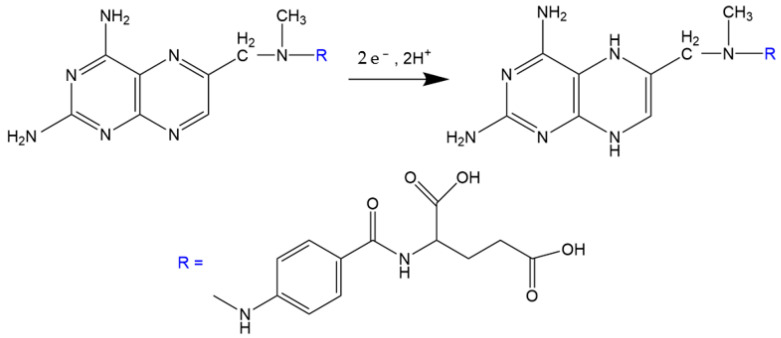
Possible mechanism involving the electroreduction of methotrexate.

**Figure 5 biomimetics-08-00077-f005:**
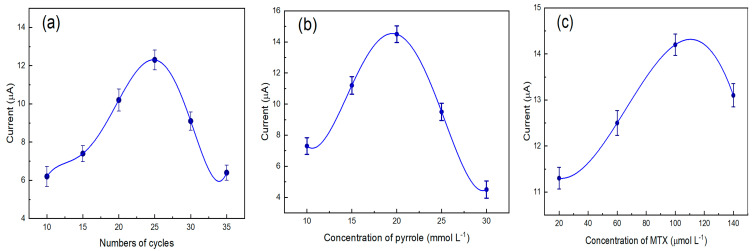
Effects of electropolymerization of pyrrole on MWCNT/GCE: Number of cycles (**a**), the concentration of pyrrole (**b**), and concentration of methotrexate (**c**).

**Figure 6 biomimetics-08-00077-f006:**
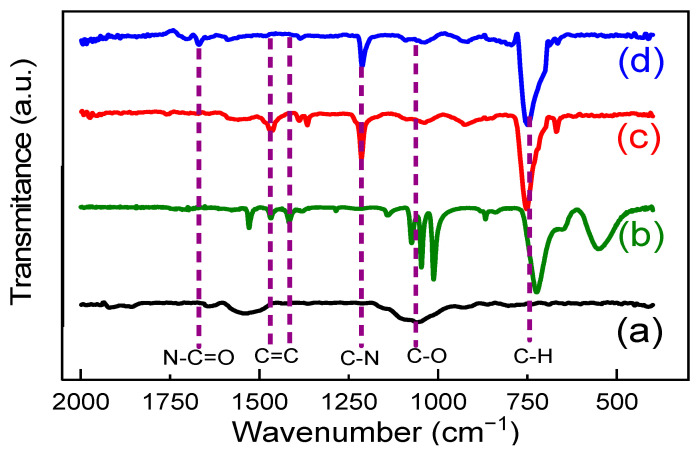
FTIR Spectrum of: (**a**) MWCNT, (**b**) pyrrole, (**c**) PPy/MWCNT, and (**d**) OPPy/MWCNT.

**Figure 7 biomimetics-08-00077-f007:**
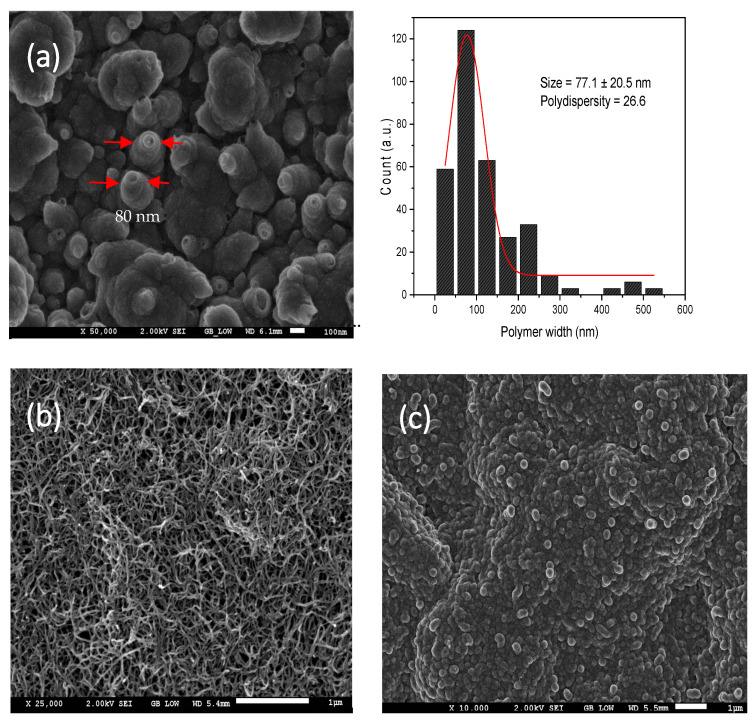
SEM characterization of MIP/GCE (**a**), MWCNT/GCE (**b**), and MIP/MWCNT/GCE (**c**).

**Figure 8 biomimetics-08-00077-f008:**
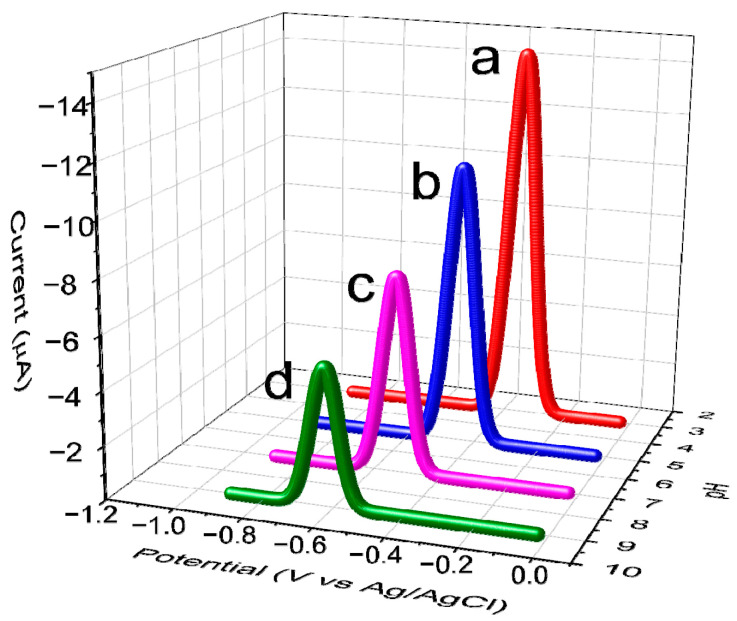
DP voltammograms obtained from the application of 100 µmol L^−1^ MTX at different pH levels of PBS. Range of pH investigated: 3 (**a**), 5 (**b**), 7 (**c**), and 9 (**d**).

**Figure 9 biomimetics-08-00077-f009:**
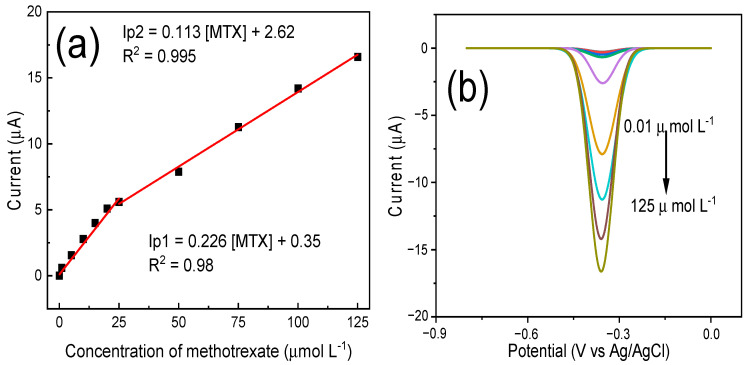
(**a**) Calibration curve for methotrexate detection based on the application of MIP/MWCNT/GCE and (**b**) differential pulse voltammogram in a concentration range of 1 × 10^−8^ to 125 × 10^−6^ mol L^−1^ using PBS at pH 3 as electrolyte solution.

**Figure 10 biomimetics-08-00077-f010:**
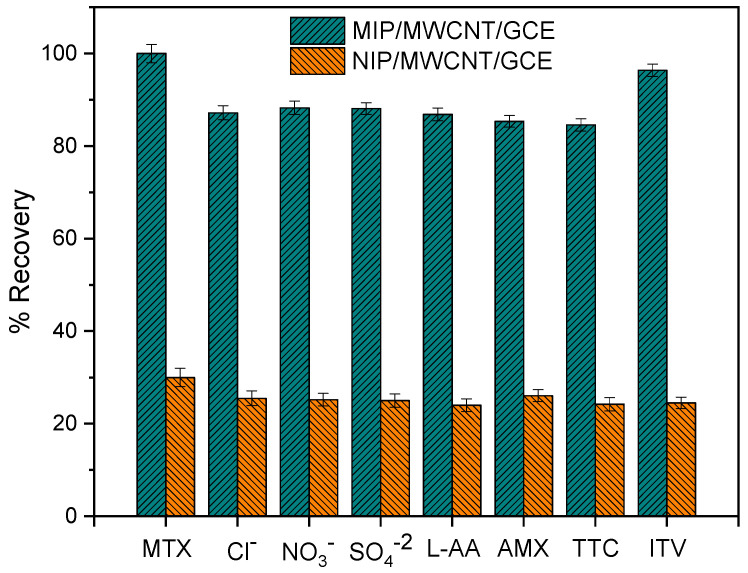
Interference study conducted using MIP/MWCNT/GCE and NIP/MWCNT/GCE.

**Table 1 biomimetics-08-00077-t001:** Results obtained from the analysis of selectivity of MIP/MWCNT/GCE and NIP/MWCNT/GCE in terms of the quantification of methotrexate in the presence of interfering molecules.

Molecule	Recovery MIP (%)	Recovery NIP (%)	α ^(1)^	β ^(2)^
MTX	100	30	3.33	1.00
Cl^−^	87.2	25.5	3.42	0.87
NO_3_^−^	88.29	25.2	3.50	0.88
SO_4_^2−^	88.1	25	3.52	0.88
L−AA	86.83	24	3.62	0.87
AMX	85.35	26	3.28	0.85
TTC	84.58	24.2	3.50	0.85
IVT	96.4	24.5	3.93	0.96

(1) α = I_MIP_/I_NIP_, (2) β = I_MIP_/I_Interference_.

**Table 2 biomimetics-08-00077-t002:** Results obtained from the analysis of river water samples.

Sample	[Methotrexate]/mol L^−1^			Recovery (Sensor, %) **	Error *** (%)
	Added	Found *			
		Proposed Method	HPLC Method		
**Water River**	2.50 × 10^−6^	(2.61 ± 0.1) × 10^−6^	(2.55 ± 0.07) × 10^−6^	102	2.35
	2.50 × 10^−5^	(2.64 ± 0.08) × 10^−5^	(2.59 ± 0.04) × 10^−5^	104	1.93

* *n* = 3; ** Recovery = [(Found concentration)/(Added concentration)] × 100%; *** Error = [sensor method concentration) − (HPLC method concentration)/(HPLC method concentration)] × 100%.

**Table 3 biomimetics-08-00077-t003:** Results obtained from the analysis of pharmaceutical formulation samples.

Sample	Declared Value (mg)	Methotrexate per Tablet (mg)			
		Proposed Method	Comparative Method	Recovery (Sensor, %) *	Relative Error (%) **
**Pharmaceutical formulation**	2.50	2.56 ± 0.09	(2.62 ± 0.03)	105	2.29

* Recovery percentage = [Found/Added] × 100; ** Relative error = [(Proposed method − Comparative method)/(Comparative method)] × 100.

## Data Availability

Not applicable.

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
