# Peer review of "Biomimetic Material for Quantification of Methotrexate Using Sensor Based on Molecularly Imprinted Polypyrrole Film and MWCNT/GCE"

_biomimetics, 2023, doi:10.3390/biomimetics8010077_

Round 1

Reviewer 1 Report

The manuscript presents the detection and quantification of an emerging environmental pollutant, methotrexate using MIP film generated from polypyrrole while employing MWCNTs to improve the conductivity hence sensitivity of the sensor. The work is interesting but its publication in the current state could not be suggested since it contains many issues worthy of attention. Some of such issues are raised below.

1. The title contains the expression “biomimetic quantification” that seems confusion. What do the authors mean by this? Moreover, the sense of the expression was not elucidated in the manuscript.

2. The introduction needs to be edited for clarity and sequential flow of thoughts. Below are some of the issues identified in this part.

3. Lines 78-79, the sentence ''To develop an effective sensor, it is crucial to choose.'' seems incomplete.

4. Why was the special interest in DNA/RNA among other natural recognition elements in the sentence ''To generate effective sensing responses, recognition elements like DNA/RNA aptamers and oligopeptides generated using display techniques must be immobilized or coated on a transducing substrate.'' when neither DNA or RNA is used in the main work.

5. The following sentence should be carefully adjusted ''Potentiometric, voltammetric, and impedance spectral sensors are examples of biomimetic sensors that have been upgraded with carefully selected biomimetic materials.''

6. Of what significance or relevance is the following statement to the preceding one? ''There can be more variation in analyte identification and signal transduction with the use of bioengineered cells.''

7. Authors should clearly enumerate the connection, if any, between biomimetic sensors and MIP based sensors. Currently, the MS shows no relationship.

8. The sequence and logic of the sentences in lines 104-115 should be checked.

9. In addition, the authors should explain how the MIP sensor will benefit from ”the enhancing the electrical conductivity of the material” (line 107).

10. The literature should also be expanded to include some records of the detection of other environmental pollutants such as antibiotics using MIP as a biomimetic recognition element. Examples include https://doi.org/10.3390/bios12070441, https://doi.org/10.1016/j.snb.2022.132768.

11. How did authors ensured the protection of the structural integrity of target during polymerization for such a wide range of synthesis potential (-0.4 to 1 V). In fact, in lines 194-196 authors stated that both methotrexate and pyrrole were simultaneously oxidized at the same potential (of +0.8 V) and attributes such observation to diffusion of methotrexate over the substrate. What is the meaning of this explanation?

12. It would be also useful to show the CV of Methotrexate in the potential range and electrolyte used for MIP synthesis.

13. Figure 2 seems to be very complicated and was not clearly described in the 3.1. Why was the accumulation potential applied at this stage? Why the electrochemical response corresponded to the potential range -0.4 to 0.9 V with Eac of 0.4 V was considered as better than others?

14. What is the difference between the accumulation potential used in 3.1 and the conditioning potential used in 3.2?

15. The explanation of the DPV curves presented in Figure 3 is not clear enough to understand the sensing principle of the MIP/MWCNT/GCE that is obviously based on the enhancing of the electrochemical reduction of MTX. The DPV curve of the MWCNT/GCE should also added to the graph in Fig. 3 for comparison. How was the electroactive area of the studied electrodes calculated? The equation is missing.

16. The sentence “As can be noted, the application of polypyrrole (which is a conductive polymer in the presence of the imprinted cavities) led to a slight increase in the electroactive area.” (lines 244-245) is confusing. Thus, the authors stated that polypyrrole is conductive in the presence of imprinted cavities. But this is not correct. Polypyrrole is a conductive polymer due to the presence of the delocalized π-electron system along the backbone and the doping occurring during the synthesis. The formation of imprinted cavities in polypyrrole does not contribute to the conductivity. In contrast, according to the template removal procedure (i.e. the formation of imprinted cavities) used in this study and consisted in the immersion of the modified electrode in buffer with pH 10 and applying potential cycles between 0.3 - 1 V , more likely resulted in the dedoping of polypyrrole and loss of conductivity.

17. Ref. 37 in the caption of Figure 4 seems not appropriate.

18. Caption of Figure 5 is confusing, „…electropolymerization of the MIP/MWCNT/GCE“ is not correct.

19. Caption of Figure 6 is confusing, does not describe the presented spectra correctly.

20. The description of the FTIR spectra (lines 271-276) is inadequate. It should be considerably improved.

21. In figure 7, authors should label the part showing the size and polydispersity with an appropriate caption.

22. How was the size and polydispersity measured?

23. Overall, the discussion provided on the morphology of the synthesized polypyrrole (lines 277-291) seems confusing. It is not clear why the potential applied before electrodeposition stimulated „the vertical growth“ of the polymer.  Which picture in Fig. 7 corresponds to the polypyrrole nanotubules?

24. The purpose of introducing the paragraph in lines 292-300 is not clear.

25. What is the pH of the environmental media in which MTX is to be detected? What is the effect of such pH to the optimal pH of 3 at which maximum adsorption is obtained?

26. Lines 399-400. The sentence “The sensor was prepared by taking advantage of the methotrexate imprinted cavities derived from the electropolymerization of pyrrole on GCE modified with MWCNT.” Is confusing.

Author Response

1st Reviewer's Comments

            Highlight in yellow color

The manuscript presents the detection and quantification of an emerging environmental pollutant, methotrexate using MIP film generated from polypyrrole while employing MWCNTs to improve the conductivity hence sensitivity of the sensor. The work is interesting but its publication in the current state could not be suggested since it contains many issues worthy of attention. Some of such issues are raised below.

Thanks for your important comments and we acknowledge the referee’s time and work dedicated to our manuscript.

Question (1) the title contains the expression “biomimetic quantification” that seems confusion. What do the authors mean by this? Moreover, the sense of the expression was not elucidated in the manuscript.

Answer (1) New title has been added.

Question (2) The introduction needs to be edited for clarity and sequential flow of thoughts. below are some of the issues identified in this part

Answer (2) The Introduction has been edited as suggested

Question (3) Lines 78-79, the sentence ''To develop an effective sensor, it is crucial to choose.'' seems incomplete.

Answer (3)   The Sentence has been completed

Question (4) Why was the special interest in DNA/RNA among other natural recognition elements in the sentence ''To generate effective sensing responses, recognition elements like DNA/RNA aptamers and oligopeptides generated using display techniques must be immobilized or coated on a transducing substrate.'' when neither DNA or RNA is used in the main work.

Answer (4) Explanation has been added

Question (5) The following sentence should be carefully adjusted ''Potentiometric, voltammetric, and impedance spectral sensors are examples of biomimetic sensors that have been upgraded with carefully selected biomimetic materials.''

Answer (5) The sentence has been adjusted carefully

Question (6) Of what significance or relevance is the following statement to the preceding one? ''There can be more variation in analyte identification and signal transduction with the use of bioengineered cells.'

Answer (6) Now it is corrected

Question (7) Authors should clearly enumerate the connection, if any, between biomimetic sensors and MIP based sensors. Currently, the MS shows no relationship.

Answer (7) it has been corrected. In this workd, the MIP is the biomimetic material.

Question (8) The sequence and logic of the sentences in lines 104-115 should be checked.

Answer (8)   it has been checked and corrected

Question (9) In addition, the authors should explain how the MIP sensor will benefit from “the enhancing the electrical conductivity of the material” (line 107).

Answer (9) the future applications and prospects has been added to the main text

Question (10) The literature should also be expanded to include some records of the detection of other environmental pollutants such as antibiotics using MIP as a biomimetic recognition element. Examples include https://doi.org/10.3390/bios12070441, https://doi.org/10.1016/j.snb.2022.132768

Answer (10) It has been added

 Question (11) How did authors ensured the protection of the structural integrity of target during polymerization for such a wide range of synthesis potential (-0.4 to 1 V). In fact, in lines 194-196 authors stated that both methotrexate and pyrrole were simultaneously oxidized at the same potential (of +0.8 V) and attributes such observation to diffusion of methotrexate over the substrate. What is the meaning of this explanation?

Answer (11) has been corrected. Thanks for the important question. As emphasized, at a potential of 0.8 V, oxidation of the analyte and polypyrrole occurs. The result is the peak current obtained at this potential. All oxidized analyte will be reduced at the -0.3V potential, and if there was no analyte on the electrode surface during pyrrole electropolymerization, no current signal would be observed.

Question (12) It would be also useful to show the CV of Methotrexate in the potential range and electrolyte used for MIP synthesis.

Answer (12) See the figure 1

Figure 1. Cyclic voltametr for (a) GCE y (b) MWCNT/GCE in the standard solution of metotrexato 100 µ molL-1, ʋ = 50 mVs-1.

Question (13) Figure 2 seems to be very complicated and was not clearly described in the 3.1. Why was the accumulation potential applied at this stage? Why the electrochemical response corresponded to the potential range -0.4 to 0.9 V with Eac of 0.4 V was considered as better than others?

Answer (13) Now it has been explained in detail in the manuscript, please see Figure 2 also.

Question (14) What is the difference between the accumulation potential used in 3.1 and the conditioning potential used in 3.2?

Answer (14) Now it has been explained in detail in the manuscript, See Figure 2 now it is clear

Question (15) The explanation of the DPV curves presented in Figure 3 is not clear enough to understand the sensing principle of the MIP/MWCNT/GCE that is obviously based on the enhancing of the electrochemical reduction of MTX. The DPV curve of the MWCNT/GCE should also added to the graph in Fig. 3 for comparison. How was the electroactive area of the studied electrodes calculated? The equation is missing.

Answer (15) It has been explained in detail.

Question (16) The sentence “As can be noted, the application of polypyrrole (which is a conductive polymer in the presence of the imprinted cavities) led to a slight increase in the electroactive area.” (lines 244-245) is confusing. Thus, the authors stated that polypyrrole is conductive in the presence of imprinted cavities. But this is not correct. Polypyrrole is a conductive polymer due to the presence of the delocalized π-electron system along the backbone and the doping occurring during the synthesis. The formation of imprinted cavities in polypyrrole does not contribute to the conductivity. In contrast, according to the template removal procedure (i.e. the formation of imprinted cavities) used in this study and consisted in the immersion of the modified electrode in buffer with pH 10 and applying potential cycles between 0.3 - 1 V, more likely resulted in the dedoping of polypyrrole and loss of conductivity.

Answer (16) it has been explained in detail.

Question (17) Ref. 37 in the caption of Figure 4 seems not appropriate

Answer (17) Ref 37 has been deleted

Question (18) Caption of Figure 5 is confusing, „…electropolymerization of the MIP/MWCNT/GCE“ is not correct.

Answer (18) Now it has been corrected

Question (19) Caption of Figure 6 is confusing, does not describe the presented spectra correctly.

Answer (19) It has been corrected

Question (20) The description of the FTIR spectra (lines 271-276) is inadequate. It should be considerably improved.

Answer (20) it is improved

Question (21) In figure 7, authors should label the part showing the size and polydispersity with an appropriate caption.

Answer (21) it has been corrected as suggested

Question (22) How was the size and polydispersity measured?

Answer (22) now it is given in detail in the text

Question 23 Overall, the discussion provided on the morphology of the synthesized polypyrrole (lines 277-291) seems confusing. It is not clear why the potential applied before electrodeposition stimulated, the vertical growth” of the polymer.  Which picture in Fig. 7 corresponds to the polypyrrole nanotubules?

Answer (23) In other research works, it is shown that polypyrrole has a characteristic morphology called "cauliflower", this morphology is due to polymeric growth in position a and β. Therefore, in the case of applying a potential as a pretreatment, it should be expected that the morphology changes and has a horizontal or vertical growth, depending on the applied potential as given in figure 2 and 3 below.

Figure 2.  Influence of conditioning potential prior to the electropolymerization of pyrrole.

. pyrrole molecule possesses resonance structures

vertical growth

It is very probable that a resonant pyrrole molecule has been attracted as described in Figure 3 and then, when carrying out the electropolymerization, a growth of the polymer occurs in position a, managing to form a vertically growing polymer.

Figure 3. Proposed oriented electropolymerization process by applying conditioning potential before polymerizing.

Question (24) The purpose of introducing the paragraph in lines 292-300 is not clear.

Answer (24) The paragraph was corrected.

Question (25) What is the pH of the environmental media in which MTX is to be detected? What is the effect of such pH to the optimal pH of 3 at which maximum adsorption is obtained?

Answer (25) see 3.4 and mechanism of Figure 4

Figure 4. Acid-base balance in an aqueous methotrexate solution.

Question 26 Lines 399-400. The sentence “The sensor was prepared by taking advantage of the methotrexate imprinted cavities derived from the electropolymerization of pyrrole on GCE modified with MWCNT.” Is confusing.

Answer (26) Now it has been corrected

Reviewer 2 Report

Comments to the manuscript " Biomimetic quantification of methotrexate using molecularly 2 imprinted polymer sensor film based on molecularly imprinted 3 polypyrrole and MWCNT/GCE." The manuscript showed interesting results with a comprehensive discussion of the matters. However, the authors should state more about the novelty of this work and make a better comparison with other research works. Furthermore, conclusions also must be improved. They are very superficial. Please find below the other comments:

Can the authors elaborate more on the inhibited metabolic processes and the consequences in the environment of this compound?

Can the authors elaborate more on what other studies showed in the detection of these compounds?

Include more description of how samples were prepared in the characterization methods.

How do the authors ensure the removal of the template?

Line 153, how much volume?

Figure 4; there is a typo in the NH2 bind to the molecule in the first structure.

Please, present the FTIR spectra in a better way. Change the x scale from 4000-500 cm^-1. Indicate the functional groups on the spectra, and make them bigger. Make a better description of the FTIR spectra.

Can the authors present some of the SEM at higher magnifications?

Authors must include the sample collection and the physicochemical properties of the water samples. Did the authors make any pretreatment of the samples?

In the Quantification of methotrexate in pharmaceutical and river water samples are the results statistically significant? I will mention that the authors are validating the results with HPLC. The HPLC conditions are missing in materials and methods.

Line 318, if the maximum adsorption capacity was achieved at pH3, are the authors adjusting the pH in their experiments? What pH are they using?

Author Response

2nd Reviewer's Comments highlight in green color

Comments to the manuscript " Biomimetic quantification of methotrexate using molecularly 2 imprinted polymer sensor film based on molecularly imprinted 3 polypyrrole and MWCNT/GCE." The manuscript showed interesting results with a comprehensive discussion of the matters. However, the authors should state more about the novelty of this work and make a better comparison with other research works. Furthermore, conclusions also must be improved. They are very superficial. Please find below the other comments:

Thanks for your important comments and we acknowledge the referee’s time and work dedicated to our manuscript.

Question (1) Can the authors elaborate more on the inhibited metabolic processes and the consequences in the environment of this compound?

Answer (1) The main target of methotrexate is the DHFR enzyme (see Figure 5). Inhibition of DHFR leads to partial depletion of the FH4 cofactors (5-10 methylene tetrahydro folic acid and N-10 formyl tetrahydro folic acid) necessary for the respective synthesis of thymidylate and purines. In addition, methotrexate, like its physiological counterparts (folates), is converted to a series of polyglutamates (MTX-PG) in normal and tumor cells.

Figure5. Sites of action of methotrexate.

(Brunton, Hilal-Dandan, and Knollmann 2018)

Question (2) Can the authors elaborate more on what other studies showed in the detection of these compounds?

Answer (2) Table S1 shows others related studies.

Question (3) Include more description of how samples were prepared in the characterization methods.

Answer (3) To carried out the characterizations the no preparation of the samples was make.

Question (4) How do the authors ensure the removal of the template?

 Answer (4) it is explained in detail

Question (5) Line 153, how much volume?

Answer (5) It has been added in the text

Question (6) Figure 4; there is a typo in the NH2 bind to the molecule in the first structure.

Answer (6) It is given in the Figure 4

Question (7) Please, present the FTIR spectra in a better way. Change the x scale from 4000-500 cm-1. Indicate the functional groups on the spectra, and make them bigger. Make a better description of the FTIR spectra.

Answer (7) Please see the reply of reviewer 1, Questions 19 and 20.

Question (8) Can the authors present some of the SEM at higher magnifications?

Answer (8) Please see the following images

Question (9) Authors must include the sample collection and the physicochemical properties of the water samples. Did the authors make any pre-treatment of the samples?

Answer (9) detail are given in the text water were used directly and here was no pre-treatment of water samples.

Question (10) In the Quantification of methotrexate in pharmaceutical and river water samples are the results statistically significant? I will mention that the authors are validating the results with HPLC. The HPLC conditions are missing in materials and methods.

Answer (10) it is given in Table 3

Question (11) Line 318, if the maximum adsorption capacity was achieved at pH 3, are the authors adjusting the pH in their experiments? What pH are they using?

Answer (11) added HCl 0.05 mol L-1 to obtained a pH 3 also See Figure 8.

Round 2

Reviewer 1 Report

The manuscript was improved, but still it needs revisions.  Introduction in whole still not very consistence and tries to cover all possible variations of biomimetic sensors. I would suggest to rewrite it putting more focus on MIP sensor with electrochemical detection. 

1. Lines 101-102. Why did the author take these particular references? There are many examples of protein-MIP, but if the matter was low electrical conductivity which could be a problem for if electrochemical detection method was used, then I suggest to focus here on recent electrochemical protein-MIP papers e.g.: doi:10.1016/j.talanta.2022.123737, doi:10.1016/j.snb.2021.131160, doi:10.1016/j.bios.2021.113029.

2. In response to Q5. L83-84: “Some sensors of detection like potentiometric, voltammetric, and impedance have been upgraded with selected biomimetic materials.” Still sounds strange. Detection of what?

3. In response to Q5. I still see no improvement. May be this sentence can be deleted? 

4. In response to Q7. Yes, I know that MIPs is considered as a biomimetic material but it is still not well discovered in MS.

5. In response to Q8/Q9. I still do not see logic. What do the authors concern, conductivity of substrate or a MIP? How do the authors suggest to resolve it, introducing MWCNT and using of polypyrrole to build a MIP? The text does not explain this clearly. Also, still not clear how the MIP sensor will benefit from “the enhancing the electrical conductivity of the material”, since the authors answer is “Answer (9) the future applications and prospects has been added to the main text”.

6. In response to Q11. I do not understand how the answer answers the question. Moreover, if I look Fig. S2a and S2b I see that there are no much differences. 

7. In response to Q12. I did not find the answer in the suggested Fig. 1. 

8. In response to Q13. The author should discuss why -0.4 to 0.9 V gives better conductivity than 1V. Still it is not clear why it would be better for MIP. Moreover, the data points in Fig. 2 should not be connected since these curves do not show dependence.  

9. In response to Q15. The authors have not added DPV curve of the MWCNT/GCE as requested, so I cannot judge the answer. 

10. In response to Q25. The authors have not answered “What is the pH of the environmental media in which MTX is to be detected?”.

11. In response to Q26. The sentence “The sensor that was prepared was based on the technology of MIPs and thus obtain the selective printed cavities to methotrexate through the electropolymerization of pyrrole in GCE modified with MWCNT.” Is getting more confused.

Author Response

Please, see the responses to Reviewer 1 in the attach file.

Reviewer 2 Report

The authors improved the quality of the paper, and they also answered all my comments.

Author Response

Dear Reviewer 2, thank you so much foo your response